# Rational Design of Adenylate Kinase Thermostability through Coevolution and Sequence Divergence Analysis

**DOI:** 10.3390/ijms22052768

**Published:** 2021-03-09

**Authors:** Jian Chang, Chengxin Zhang, Huaqiang Cheng, Yan-Wen Tan

**Affiliations:** 1State Key Laboratory of Surface Physics, Multiscale Research Institute of Complex Systems, Department of Physics, Fudan University, Shanghai 200433, China; 13110190056@fudan.edu.cn (J.C.); huaqiangch@gmail.com (H.C.); 2School of Life Science, Fudan University, Shanghai 200433, China; zcx@umich.edu; 3Department of Computational Medicine and Bioinformatics, University of Michigan, Ann Arbor, MI 48109, USA

**Keywords:** protein engineering, thermal stability, primary sequence, sequence profile comparison, adenylate kinase

## Abstract

Protein engineering is actively pursued in industrial and laboratory settings for high thermostability. Among the many protein engineering methods, rational design by bioinformatics provides theoretical guidance without time-consuming experimental screenings. However, most rational design methods either rely on protein tertiary structure information or have limited accuracies. We proposed a primary-sequence-based algorithm for increasing the heat resistance of a protein while maintaining its functions. Using adenylate kinase (ADK) family as a model system, this method identified a series of amino acid sites closely related to thermostability. Single- and double-point mutants constructed based on this method increase the thermal denaturation temperature of the mesophilic *Escherichia coli* (*E. coli*) ADK by 5.5 and 8.3 °C, respectively, while preserving most of the catalytic function at ambient temperatures. Additionally, the constructed mutants have improved enzymatic activity at higher temperature.

## 1. Introduction

Protein thermostability improvement is one of the most essential and sought after protein engineering aspects [1]. Increasing the thermostability of an enzyme can boost the total turnover number and extend the shelf life of derived products [2]; thermostable proteins in drugs or vaccines can reduce the cost of storage and transportation [3]; high thermostability can help crystalize unstable proteins and thus determine their structures [4]. Directed evolution and rational design are two approaches for protein engineering. Thermal stability improvement is one of the many tasks that directed evolution can readily accomplish. However, directed evolution can be laborious and costly [5]. By contrast, rational design shows the extent to which we understand the functioning principles of proteins. When applied properly, rational design can be time-saving and lower the barrier for thermal stability engineering.

Most of the stability rational design methods require protein tertiary structural information. Structural information can be used to calculate surface charge distribution, backbone torsion, spatial distance between residues, side chain volumes, and solvent accessibility for improved thermostability [6,7,8]. When the protein structure is unavailable, options for effective protein engineering are limited. Among few sequence-based stability engineering methods, the improved configurational entropy (ICE) algorithm required only primary sequences of homologous proteins for thermostability improvement design through minimizing the local structural entropy of the secondary structure [9,10]. However, because of the lack of consideration for long-range side-chain contacts of residues that are far apart along the primary sequence, ICE might stabilize local structures at the cost of interrupting other noncovalent interactions such as favorable ion pairs, hydrogen bonds, or hydrophobic interactions that link distant parts of the protein backbone [11]. Consensus method hypothesized that residues contributing to catalysis or stability were more likely to be conserved during the evolution [1]. This approach was operated by replacing targets’ residue with the most frequent one among homologues. It has the advantage of screening for improved thermal stability in a single step with just sequence information. Unfortunately, after multiple residue substitutions, consensus proteins may not retain their biological activity or be unable to fold accurately [12]. In the consensus method, the correlation among residue pairs are disregarded, especially for poorly-conserved sites [1]. In addition, the consensus method usually ignores the difference between mesophiles and thermophiles, which may lead to biased results. For example, a consensus method may favor an amino acid type due to its high frequency among mesophilic proteins in the family, even though the residue type that actually stabilizes the protein may be a different amino acid type found in a smaller number of thermophiles.

Nature has presented abundant evolutionary traces with potential clues for thermal stability engineering. Thermophiles living in extremely hot environments have proteins that are largely homologous to that of mesophiles. These homologous proteins execute the same function as their mesophilic counterparts, although with different sequences and considerably higher working temperatures. However, using thermophilic homolog sequences directly for thermostability improvement would result in poor activity at ambient temperature.

In this study, we designed a novel primary-sequence-based bioinformatics method to generate thermostable variants from mesophilic enzyme, *E. coli* ADK. ADK is a phosphotransferase enzyme involved in catalyzing the interconversion of adenine nucleotides—adenosine triphosphate (ATP), adenosine monophosphate (AMP), and adenosine diphosphate (ADP). A number of previous studies have provided various results for thermostability improvement of this model protein [7,9,13,14,15]. Therefore, we had chosen mesophilic *E. coli* ADK as the engineering target to examine our method. Our primary-sequence-based bioinformatics method could identify residues that are crucial for structural stability without drastically changing their functionality. Firstly, we applied the residue correlation analysis (RCA) coevolution algorithm to analyze evolutionary information embedded in multiple sequence alignment (MSA) to detect thermostability-related residues [16]. Secondly, protein sector method proposed by Halabi et al. was applied to categorize residues according to their intrinsic correlations [17]. Then, amino acid compositions of thermophile and mesophile were compared to identify the sector that are related to structural stability. Lastly, residues within the chosen sector were mutated to amino acids suggested by the thermophilic sequence profile. In this work, single- and double- point mutations by replacing original residues with amino acids having different chemical properties were characterized and achieve up to 5.5 and 8.3 °C higher melting temperature (*T_m_*) than that of the wild type, which successfully validated our method.

## 2. Results

### 2.1. Protein Sectors Derived from Amino Acid Sequences

The foundation of our method is an MSA of the protein of interest. Through sequence alignment, a 9203 × 214 matrix of amino acid distribution was obtained within the ADK protein family, where 9203 and 214 are the number of homologous sequences and the number of amino acid positions in *E. coli* ADK, respectively. The details about the MSA of ADK family were displayed in the Appendix A. This MSA contained the information of residue transfer at corresponding sites in species evolution and was used as the input of the RCA, which outputs a 214 × 214 matrix for the evolutionary coupling strength of all residue pairs (Figure 1A).

To reduce the noise in the RCA, eigen-decomposition is performed to generate 214 eigenvalues λ_1_, λ_2_, … λ_214_ (ranked in the descending order of value). Among them, λ_1_, λ_2_, λ_3_, and λ_4_ were considerably larger than the remaining eigenvalues. According to the protein sector method [17], the eigen-decomposition components with eigenvalues lower than those generated from a randomized RCA can be attributed to a limited sampling of sequences (statistical noise). Furthermore, the largest component, λ_1_, arises from the phylogenetic effect within the ADK family (historical noise). After removing all noises, the raw $r_{ij}$ value for residue pair *i* and *j* can be approximated as the sum of three eigen-decomposition components related to λ_2_, λ_3_, and λ_4_. The mode 2–4 in eigen-decomposition contain information critical for the structure and function of ADK and can be used to approximate the global correlation matrix between the residues of the ADK family (See Appendix A).
(1)$$r_{ij}\approx \sum_{k=2}^{4}v_{k}\lambda_{k}v_{k}^{T}$$

According to protein sector method [17], in the three-dimensional eigenspace representation formed by three modes, residues with similar characteristics will gather into clusters. In our case, all 214 ADK residues can be plotted in this eigenspace (Figure 1B). After eliminating residues (plotted as the gray plus signs) with eigenvalues smaller than the boundary set by randomized RCA eigenvalues, the remaining residues can be classified into four clusters using k-mean clustering (Figure 1B), which are plotted respectively in green, orange, blue, and magenta. Among them, the green and orange sectors reveal obvious continuity in primary sequence while residues in the blue and magenta sectors locate on the whole sequence in a dispersed manner (Figure 1C).

### 2.2. Thermostability-Related Sector Identification and Mutant Design

To determine sites that should be mutated, sequence entropy analysis was applied using the subset of 256 and 3535 homologous ADK protein sequences which were confirmed to be encoded by thermophilic and mesophilic organisms according to credible published literatures or databases as listed in the Appendix A. The remaining subset of 5412 out of all 9203 sequences were not associated with information on the optimal growth temperatures of their host organisms and therefore excluded from the sequence entropy analysis. The details about how to classify the thermophilic and mesophilic species were exhibited in the Appendix A. For each position, a pair of relative entropy vectors can be obtained, which contained the information of amino acid constitutions of thermophilic and mesophilic proteins. The relative entropy angle *θ* represents the difference between the two sequence profiles. Intuitively, the greater the *θ*, the more divergent was the amino acid composition at that position of two profiles. In this study, sites with low conservation scores but high *θ* values were chosen for further mutation because highly conserved positions were crucial for preserving molecular function, whereas we hypothesized that sites critical for stabilizing the molecular structure were highly differentiated during adaptation to various environments.

Four sectors were classified based on an RCA, but not all of them might be correlated with ADK thermal stability. To determine which sector might be the most relevant to protein’s stability, we combined results obtained from the RCA and the relative entropy angle *θ* from sequence entropy analysis and finally chose the magenta sector to be mutated for further thermal stability improvement. The reason is the following: among all four sectors, the magenta one had the maximum average *θ* value (0.4727) and the blue one had the minimum average *θ* value (0.1342). Furthermore, both green and orange sectors were located closely along the one-dimensional sequence, suggesting that they could form connected domain or subdomain structures and play essential roles in the catalytic function. To examine how mutations in green, blue, and orange sectors influence the enzymatic function and protein stability of ADK, we have performed control experiments on the mutants from these three sectors. We picked two mutations from each sector according to the same procedure as in the magenta sector. Six mutants—Q18K, F19R, S43T, M96L, R131A, and V148K were generated and characterized. Their corresponding assay and circular dichroism (CD) experiment results were displayed in the Appendix A (See Appendix A), which indicated that the blue sector was highly conserved for ADK’s enzymatic function and had little to do with structure stability while the green and orange sectors might be both responsible for the enzymatic activity and protein thermostability. Therefore, they were excluded from subsequent considerations.

Combining the RCA and sequence entropy analysis, residues in the magenta sector with the eight highest relative entropy angle *θ* (least conserved) were selected for mutations (Figure 1C). Up to now, we have already recognized those protein sites responsible for thermal stability but it remained unclear which amino acids should replace the original ones. In our initial attempt, residues at those chosen sites were mutated to the corresponding amino acids that were the most frequent in thermophile ADK MSA (Figure 1D). If the same amino acid was the most frequent in both thermophile and mesophile ADK, we moved to the second most frequent one. Eight mutants of *E. coli* ADK (ADK wt) were thus generated: S41K, D76V, G100N, I101R, P139K, P140S, K141R, and R206F.

### 2.3. Thermostability Characterization of Mutants

We examined the thermostability variations of the eight mutants with CD T–melt measurements. The results showed that, compared with the *T_m_* of the wild type, four of the eight mutants—G100N, P139K, P140S, and R206F—had *T_m_* increase of 1.8 °C, 2.3 °C, 1.8 °C, and 5.5 °C, respectively (Table 1). However, the others, S41K, D76V, I101R, and K141R, had their *T_m_* decrease by 0.7–5.4 °C and were therefore mutated to the next most common residues in the thermophile: S41A, D76N, I101T, and K141P. What’s more, compared with the first four mutants (S41K, D76V, I101R, K141R), all new residues in second mutations (S41A, D76N, I101T, K141P) had side chains with apparently different properties. For example, lysine (K) in S41K (first mutation) was positively charged, whereas alanine (A) in S41A (second mutation) held a hydrophobic side chain. CD results of S41A, D76N, I101T, and K141P exhibited consistent *T_m_* rises: 5.1 °C, 0.9 °C, 2.3 °C, and 2.6 °C (Table 1). Considering all successful mutation cases (G100N, P139K, P140S, R206F, S41A, D76N, I101T, K141P), we suggested that the strategy for choosing mutation amino acid might be by replacing the original residues by amino acids with apparently different properties. However, this condition was optional rather than mandatory. In our amino acid selection strategy, the amino acid distribution of thermophilic sequences was the major consideration. In our primary-sequence-based method, the lack of 3D protein structural information limited our recognition on whether selection of particular amino acid type would work well or not. In future work, our method might be combined with 3D structural information to improve the accuracy of single-point mutation. In summary, we have obtained thermal stable ADK variants for all eight sites as the first step for the proof of principle for our method (Figure 2A).

As a bi-substrate enzyme involved in catalyzing the reversible phosphoryl transfer reaction: $Mg^{2+}\cdot ATP+AMP\overset{ADK}{↔}Mg^{2+}\cdot ADP+ADP$, ADK follows a random bi bi reaction mechanism [18]. However, for *E. coli* ADK, the reaction curve of varying AMP at constant ATP does not follow Michaelis–Menten kinetics. Previously, such a trend has been attributed to an AMP inhibition effect. However, in our earlier study, we have found that it was originated from a magnesium activation effect [19]. Since the overall enzymatic model is rather complicated, we will use the forward turnover rate at a representative ATP, AMP, and Mg^2+^ concentrations instead of the commonly used catalytic rate, k_cat_, to represent the engineered ADKs’ catalytic function in the current study.

In the enzymatic forward activity assay at room temperature (Figure 2B), these thermostable mutants had turnover rates at [ATP] = [AMP] = 1 mM that were 56–98% of that of the wild type among which the most stable variant, R206F, retained 75% of the activity (Appendix A). For the mutants with the lowest turnover rates, I101T and K141P, they seemed to have lost their magnesium activation effect [19] instead of losing the overall enzymatic activities. Since the optimal reaction temperatures of mutants are shifted as shown in later, the room temperature reaction rate of the mutants are not fully reflective of their activities. Nonetheless, the preservation of room temperature activity shown in Figure 2B is still an important metric to confirm the success of enzyme design, as our goal is to improve the thermal stability of the enzyme without significantly jeopardizing its activity at its original working temperature.

To examine whether the stabilizing effect of single-point mutation is additive, we then constructed four double mutants based on the template R206F because it had the maximum *T_m_* among all single-point mutants. S41A, G100N, P139K, and P140S were chosen as the second mutation sites since they not only had relatively high *T_m_* s but also preserved better enzymatic activities (Figure 3A). Finally, four double mutants R206F/S41A, R206F/G100N, R206F/P139K, and R206F/P140S were constructed and characterized. Compared with R206F, all four mutants presented increased *T_m_* but the increments are lower than the linear summation of ∆*T_m_*s from two corresponding single mutants (Figure 3C). R206F/S41A was the most prominent double-mutation variant with the highest *T_m_* rise of 8.3 °C (Table 1). However, almost all double mutants showed further decrease in enzymatic activities compared with respective single mutants (Appendix A).

Although the aforementioned experiments separately characterized the catalytic activity at ambient temperature and thermostability, we shall investigate whether and how the activity of our mutants differs from naturally occurring ADK at medium to high temperature. To this end, we performed temperature-dependent enzymatic activity experiments shown in Figure 3B, where the turnover rate profiles at [ATP] = 2.5 mM and [AMP] = 2.5 mM of ADK wt and *Aquifex aeolicus* ADK (ADK aq) showed that they had the maximum activity at approximately 50 °C and 90 °C, respectively. To prevent inaccurate enzyme activity measurement due to insufficient substrates, we increased the concentrations of ATP and AMP to 2.5 mM. The result was consistent with the results from a previous study [13]. At the industrial level, most enzymes worked at high temperatures (approximately 60 °C) where both ADK wt and ADK aq would lose the majority of their enzymatic function [20]. In our design, two selected variants, the most stable single mutant R206F and double mutant R206F/S41A, showed shifted peak catalytic activities at elevated temperatures (58–65 °C) compared with mesophilic and thermophilic natural enzymes, which validates our thermal stability modification.

## 3. Discussion

In this work, we used the RCA-based method to analyze evolutionary differences in the MSA of thermophiles and mesophiles of a model enzyme ADK. Following the concept of protein sectors [17], we isolated the group of residues that might be related to structural integrity and stability (See the magenta sector in Figure 4A). In the eigenspace of the RCA, we calculated the inner product of the vector formed through thermophilic and mesophilic MSAs to derive the relative entropy angle *θ*. This result revealed residues that are the most different between thermophiles and mesophiles. We chose eight residues with the highest relative entropy angle *θ* values, namely S41, D76, G100, I101, P139, P140, K141, and R206, and constructed eight thermally stable single-point mutants. The maximum *T_m_* increment of favorable single-point mutations can be as high as 5.5 °C. Meanwhile, the enzymatic activity of the most stabilizing mutant retains 75% of the performance at [ATP] = [AMP] = 1 mM compared with the wild-type ADK.

The prediction of the magenta sector using our primary-sequence-based method could be additionally justified by two structural aspects since the structure of ADK has been solved. First, only the magenta sector had no residues directly participating in substrate binding (see Figure 4B), which minimized the possibility of hindering the enzymatic function. In addition, results (Appendix A) showed that except for K57, V132, and K200, residues in direct contact with substrates usually had a small average *θ* value (0.0141), indicating their high evolutionary conservation. Second, the residues in the magenta sector were mainly located at the ends or turns of secondary structure elements (see Figure 4C). From this point of view, the extra stability obtained through our bioinformatics method might result from the stabilization effect of these residues on helices and sheets of ADK. It provides a clue to further optimize this algorithm by considering secondary structure information to and leading to a stabilized overall structure. Nonetheless, experimental structure determination will be necessary to exactly confirm the mechanisms behind the stabilization effect of these residues.

With thermostability elevation, enzymatic activity loss from mutants was also observed. This is a common issue in previous ADK design attempts [15,21,22]. In our case, we attributed this to that *E. coli* ADK requires a large magnitude of conformational change for substrates to enter into the reaction center and release of product [23]. Thus, mutations that stabilized the backbone also made it less flexible to accommodate the conformational change, hindering substrate turnover.

So far, our work revealed that appropriate single-point mutations could generate thermostable ADK mutants. However, multiple residue substitutions might be able to further improve the thermostability. For example, Euiyoung Bae redesigned the ADK from *Bacillus subtilis* by mutating as many as 26 residues and finally obtained a thermally stable variant with a Δ*T_m_* of 12.5 °C [9]. Moreover, different thermostability-improved mutants, which were derived from various stabilization methods, such as directed evolution, comparative study, or theoretical computation, could be combined together to form one construct with a higher *T_m_* [21]. Taking all above cases into account, we went beyond single residue substitution and constructed a set of double mutants using the eight single-mutated variants. This was to examine, through our method, whether mutants with more point substitutions could achieve a greater thermal stability gain and show additive effects in *T_m_*. As expected, four double mutants (R206F/S41A, R206F/G100N, R206F/P139K, and R206F/P140S) showed improved thermal stabilities compared with their original single-point mutants, which confirmed the synergistic effect of two stabilizing single mutations. In addition, all double mutants displayed a *T_m_* increment lower than the summation of the respective Δ*T_m_* s from the two original individual mutants. Furthermore, our results demonstrated that the additive effect weakened as the sum of two Δ*T_m_* increased (Figure 3C), which is consistent with the literature [21].

The protein sector method was previously applied to identify activity-related regions within specific proteins, but thermal stability improvement was rarely achieved. Only one report mentioned that 23 covariant residues found in GH1 β-glucosidases were confirmed using statistical coupling analysis as one protein sector. This sector could be further divided into two subsectors by using alanine-screening mutagenesis experiments, one is related to catalytic activity regulation, whereas the other is related to thermal stability [24]. Thus, the single-protein-sector architecture proposed by the previous study was too rough to accurately categorize residues. Moreover, due to the lack of statistical insight in terms of what amino acid type should a position be mutated into, mutation effects of all 23 positions were characterized by mutating into the same amino acid type (alanine). Consequently, no insight was proposed on how to read just the amino acid type for six positions whose mutation caused *T_m_* reduction. By contrast, in our study, we present a detailed protein sector assignment together with rational selection of both residue position and amino acid types for thermostable mutant construction.

Studies have shown that various techniques were complementary for the thermal stability improvement of the same protein, where different stabilizing methods generated different mutants with few overlapping sites [21]. The consensus technique considered those well-conserved sites while our method focused on the correlation in whole sequences and the divergence between mesophilic and thermophilic profile. To obtain more stable ADK mutants, integrating our method with other techniques, such as a consensus-based method, might be a promising optimization scheme for further improvements. 

Among various developed computational methods to predict how amino acid substitutions influence protein stability, the most accurate ones are still tertiary structure based [25]. Most existing programs, such as Rosetta [8], FoldX [26], and ABACUS [27], are based on the optimization of physical or statistical folding free energy functions (ΔG^0^) to determine appropriate amino acid types for different positions of given protein backbone structures. Almost all factors contributing to protein stability such as van der Waals, electrostatics, solvation, hydrogen bonding, disulfide bonding, backbone torsion, side-chain torsion and so on are totally taken into account in the construction of energy function, which guarantees reasonable accuracy of the prediction of protein modifications. However, the accuracies of these energy functions are contingent on the availability of high-resolution experimental structures of target proteins with little consideration of the dynamics of the backbone structure. These requirements limit their applications to proteins without an experimental structure. Additionally, these methods do not leverage the rich information of homologous protein sequences, which are far more readily available than experimental structures, considering only 50,000 (0.03%) of 180 million protein sequences in the UniProt database have structures experimentally solved in the Protein Data Bank database [28]. Although studies have attempted to combine the physical energy function with an additional energy term accounting for amino acid composition at individual positions of sequence profile [14,29], their profile terms lack pairwise evolutionary coupling information, which our study shows is crucial for a successful design. 

Compared with structure-based predictors, developed sequence-based methods are few in number and applied usually in the absence of experimental structure [30]. I-Mutant2.0 quantifies the ΔG^0^ change plus or minus of single residue substitution by just considering the nearest sequence neighbors when only the sequence is available [31]. It is trained and tested on a data set derived from the Thermodynamic Database for Proteins and Mutants. Then I-Mutant2.0 can be applied to any target protein. The principles of another three methods, iPTREE-STAB, MuPRO, and EASE-MM, are almost the same as I-Mutant2.0, with just different training procedures [32,33,34]. The ICE method can improve the protein thermostability by minimizing target protein’s local structural entropy in the sequence alignment of the target and another reference [9]. These five sequence-based methods present one obvious advantage of just requiring information from one or two protein sequences. However, they also have the disadvantage of just considering the local sequence information and ignoring the long-range correlation among residue pairs. In contrast, our method effectively utilizes the pairwise evolutionary coupling information embedded in the multiple sequence alignment of homologous proteins, which can be readily collected from corresponding databases. 

In summary, as our approach is solely based on evolutionary information, a future research direction may focus on a combination of evolutionary information with physical/statistical energy terms from structure-based methods. Such explicit consideration of physical interaction in the protein structure should in principle further boost the success rate of our current design efforts solely based on evolutionary signals from MSA.

## 4. Materials and Methods

### 4.1. Primary-Sequence-Based Thermostability Design

Briefly, our sequence-based protein redesign algorithm consists of the following steps.

Construct a multiple sequence alignment for the target protein family.Apply RCA to the MSA to obtain a correlation matrix.Identify protein sectors through the eigen-decomposition of the correlated matrix.Calculate the relative entropy angle *θ* from the difference between mesophiles and thermophiles. The protein sector with the largest average *θ* should be picked out for further selection of mutation sites.Replace residues whose *θ* exceed a certain threshold with the corresponding amino acids which were the most common among thermophiles or whose side chains had distinct properties.Characterize the mutants by circular dichroism measurement and enzymatic activity assay.Consider constructing double-point mutations consisting of above characterized single-point mutation sites.

Details of the whole process will be introduced step by step in the following sections.

### 4.2. Sequence Alignment and Curation

Protein sequences of the ADK family (PF00406) were obtained from Pfam database version 32.0 [35]. Sequences with domain organization labeled as “ADK, ADK_lid”, or “ADK” were retrieved. Fragments or entries that do not have the function of catalyzing the reversible phosphate group transfer between AMP and ATP according to UniProt database were removed from the pool. MSA of the remaining 9203 sequences was performed with Clustal Omega 1.2.2 using Hidden Markov Model (HMM) of PF00406 family as seed HMM [36]. The alignment output was projected onto ADK wt (P69441/KAD_ECOLI) indices such that only columns corresponding to the residues of ADK from *E. coli* were preserved. Proteins were further classified into “thermophilic”, “mesophilic”, and “unrecognized” categories according to the optimal growth temperature range of their host organisms as reported by National Center for Biotechnology Information Genome [37], UniProt and so on. Details of all data sources are listed in the Appendix A.

### 4.3. Coevolution Analysis

A modified version of the RCA was applied to locate residues related to different functions. Consider position *i* of an MSA with *M* sequences and *N* positions, the substitution event $X_{ikl}$ where the amino acid at row *k* was mutated to amino acid at row *l* could be scored using BLOSUM50 scoring matrix. If either sequence *k* or sequence *l* but not both of them contained a gap at position *i*, $X_{ikl}$ would be assigned the gap penalty −8. If both sequences *k* and *l* contained a gap, $X_{ikl}$ would be assigned the gap-to-gap score zero. Afterward, the correlation between positions *i* and *j* could be quantified using Pearson correlation coefficient between all substitution events at position *i* and all substitutions at position *j*:(2)$$r_{ij}=\left[2/\left(\left.M(M-1\right)\right)\right]\overset{M-1}{\underset{k=1}{\sum }}\overset{M}{\underset{l=k+1}{\sum }}\left[\left(X_{ikl}-\langle X_{i}\rangle \right)\left(X_{jkl}-\langle X_{j}\rangle \right)/\left(\sigma_{i}\sigma_{j}\right)\right]$$
$\langle X_{i}\rangle$ was the average for all substitution scores at position *i*, whereas $\sigma_{i}$ was the standard deviation. Thus, an *N* × *N* square matrix denoting RCA correlations could be generated from the MSA with *M* sequences and *N* positions. *N* eigenvalues λ_1_, λ_2_, …, λ_N_, ranking in a descending order, could be computed on which eigen-decomposition could be performed for the correlation matrix. The corresponding eigenvectors can be denoted as $v_{1},\;v_{2},\;\ldots ,\;v_{N},\;$ respectively.

Only the top-ranking eigen components were correlated to “real” information in the correlation matrix, while the rest were insignificant noises. Specifically, among the 214 eigenvalues λ_1_, λ_2_, …, λ_214_ (ranked in descending order of values), most components could be attributed to limited sampling of sequences (statistical noise) and the largest one corresponding to λ_1_ (13.29) arose from the phylogenetic effect within ADK family (historical noise) [17]. Therefore, only a few eigenvectors were required as an approximation for protein sector analysis of the original RCA matrix. In our work, the summation of three eigen-decomposition components related to λ_2_ (6.67), λ_3_ (4.97), λ_4_ (4.35) exhibited the features resembling the original correlation matrix (Appendix A).

The resulting three-component-approximation matrix was then used to cluster residues in the eigen-space representation. The boundary of noises determined by repeating the same calculations on a randomized ADK MSA and comparing the distributions of the two sets of eigenvalues. Specifically, we randomly shuffled the order of amino acids at each position in the alignment to eliminate all correlations between positions while preserving the amino acid frequencies at each position. 10 trials of randomization would give the average histograms of elements in eigenvector 2–4, respectively (Appendix A, bottom panels). The Gaussian fit of the three histograms could yield the positional weight boundaries, which were used to rule out the residue with weak correlation. More specifically, residues with all three position weights from eigenvector 2–4 which were smaller than the boundaries were considered to have no significant correlation (Appendix A, upper panels). We then deleted them and classified remaining residues into four clusters by K-means clustering algorithm. Finally, four sector representation of the RCA matrix was obtained.

### 4.4. Sequence Entropy Analysis

The relative sequence entropy of thermophilic/mesophilic sequence profile was used to determine residues related to thermostability. Considering position *i* at an MSA with *M* sequences, its relative entropy vector $\bar{D_{\dot{i}}}$ could be defined as:(3)$$\bar{D_{\dot{i}}}=\left[D_{i}^{a},D_{i}^{c},\cdots ,D_{i}^{y},D_{i}^{gap}\right]$$
$D_{i}^{a}$, the relative entropy of amino acid *a* at position *i*, was defined as the logarithm of binomial probability of choosing the observed number of *a* at *i*:(4)$$D_{i}^{a}=\log\left[{\left(q^{a}\right)}^{Mf_{i}^{a}}{\left(1-q^{a}\right)}^{M\left(1-f_{i}^{a}\right)}M!/\left(\left(Mf_{i}^{a}\right)!\left(M\left(1-f_{i}^{a}\right)\right)!\right)\right]\;\;\approx f_{i}^{a}\log\left(f_{i}^{a}/q^{a}\right)+\left(1-f_{i}^{a}\right)\log\left(\left(1-f_{i}^{a}\right)/\left(1-q^{a}\right)\;\right)$$
$f_{i}^{a}\;$ was the frequency of amino acid *a* at position *i*, and $q^{a}$ was the mean frequency of amino acid *a* among all proteins [17]. The “Mean frequency” of gaps $q^{gap}$ was defined as the gap frequency in the trimmed MSA. The mean frequency of all other 20 amino acids was normalized by multiplying $\left(1-q^{gap}\right)$. $\left|D_{i}^{a}\right|$ represented the difference of the probability distribution of amino acid type *a* from background probability at position *i* of the protein family. The relative entropy vectors $\vec{D_{i}}$ were separately calculated for thermophilic proteins and for mesophilic proteins. ${\vec{D^{thermo}}}_{i}$ and ${\vec{D^{meso}}}_{i}$ denoted the relative entropy vectors for the thermophilic and mesophilic sequence profiles, respectively, for the *i*th position. The angle between these two vectors represented how divergent the encoding of amino acids at position *i* was for thermophilic sequences and mesophilic sequences. We defined it as the relative entropy angle *θ*, and it can be calculated as follows:(5)$$\theta_{i}=\cos^{-1}\left[\left({\vec{D^{thermo}}}_{i}\cdot {\vec{D^{meso}}}_{i}\right)/\left(\left|{\vec{D^{thermo}}}_{i}\right|\left|{\vec{D^{meso}}}_{i}\right|\right)\right]$$

We hypothesized that residues that were essential for thermostability would score higher in this *θ* parameter. Considering the correlations between ADK residues, we selected the protein sector with largest average *θ* as the mutation target. Then residues with *θ* value above one custom threshold in this sector were mutated from their mesophilic type to the most common amino acid in that position for thermophilic sequences.

### 4.5. Materials

Tris(hydroxymethyl)aminomethane was purchased from Sangon Biotech (Shanghai) Co., Ltd. (Shanghai, China). KCl, MgOAc_2_, and bovine serum albumen (BSA) were obtained from Sinopharm Chemical Reagent Co., Ltd. (Beijing, China). ATP, AMP, and Tris(2-carboxyethyl)phosphine hydrochloride (TCEP) were obtained from BBI Life Sciences (Shanghai, China). Pyruvate kinase/lactic dehydrogenase (PK/LDH) enzymes from rabbit muscle and P^1^,P^5^-di(adenosine-5′) pentaphosphate pentasodium salt (Ap5A) were from Sigma-Aldrich(Shanghai)Trading Co.,Ltd. (Shanghai, China). Nicotinamide adenine dinucleotide disodium salt (NADH) was from Roche Diagnostics GmbH (Mannheim, Germany). Phosphoenolpyruvate (PEP) was from Alfa Aesar, Thermo Fisher Scientific (Waltham, MA, USA). All the aforementioned reagents were used without further purification.

### 4.6. ADK Purification

The wild-type ADK from *E. coli* strain K12 was used as the starting sequence in this study. The wild-type ADK and its mutants were purified in a manner similar to that described earlier [23]. The protein was first purified with a HisTrap HP histidine-tagged nickel affinity chromatography column (GE Healthcare, Chicago, Illinois, USA). Further purification was performed with a HiTrap Q HP anion-exchange chromatography column and a Superdex 200 Increase small-scale size-exclusion chromatography column (both were from GE Healthcare). The purification results of ADK wt, R206F, and R206F/S41A were shown in the Appendix A.

### 4.7. Room Temperature Forward Activity Assay of ADK

All enzymatic assays were performed with the assay buffer: 100 mM Tris (pH by acetic acid), pH 7.5, 100 mM KCl, 0.08 mg/mL (1.2 μM) BSA, and 0.6 mM TCEP. ADP production was monitored with coupling enzymes PK and LDH. Each ADP produced caused the oxidation of one NADH to NAD^+^ and was detected through the absorbance change at 340 nm [38]. The coupling reagent concentrations used were 5 mM PEP and 0.8 mM NADH with 2 μL PK/LDH (600–1000 units/mL PK and 900–1400 units/mL LDH) per 100 μL reaction. ADK’s forward reaction rates were assayed at a fixed ATP concentration, and AMP concentrations varied from 0 to 3 mM, with MgOAc_2_ concentrations at 2 mM. All the measurements were obtained in 96-well plates and read with a microplate reader (Molecular Devices SpectraMax M2) at room temperature (~25 °C). All the rates reported in this work are initial velocities, which are measured at the beginning of the reaction, when the product concentration is increasing linearly under known concentrations of enzyme and substrate. Each experiment was repeated at least three times and averaged to determine error bars. All error bars are reported to 1 standard deviation.

### 4.8. CD Temperature Melt Measurement

The $T_{m}s\;$ of wild type ADK and mutants were measured with CD spectrometer (Applied Photophysics qCD Chirascan plus, Surrey, UK). ADK sample (10 μM) in phosphate buffer (60 mM) with pH 7.4 was monitored at 222 nm to obtain the CD trajectory. The temperature range from 30 °C to 70 °C is scanned at a rate of 1 °C/min. *T_m_* is analyzed by using the modified Sigmoid function contained in the subsidiary software (CDNN2.1). All measurements were repeated three times to generate their standard deviations.

### 4.9. Temperature-Dependent Enzymatic Activity Assay

ADK wt was a mesophilic enzyme catalyzing the following reversible reaction: $Mg^{2+}\cdot ATP+AMP\overset{ADK}{↔}Mg^{2+}\cdot ADP+ADP$. Activity assays at various temperatures were implemented in the direction of ADP formation by using a modified end-point method as described previously [38]. Coupling buffer, ATP, and AMP solutions were prepared in advance. The final assay system consisted of 100 mM Tris, pH 7.5, 100 mM KCl, 1.2 μM BSA, 0.8 mM TCEP, 2 mM MgOAc_2_, 2.5 mM ATP, 2.5 mM AMP, and 5 nM ADK. Before the reaction, the mixture of 38 μL coupling buffer, 20 μL ATP, and 20 μL AMP was heated to the desired temperature for 2 min. A preheated ADK solution of 22 μL (22.72 nM in 100 mM Tris, pH 7.5, 100 mM KCl) was blended with the mixture to initiate the reaction, and it lasted for 1 min. The enzyme reaction was terminated immediately at the end of 1 min by adding 1 mM Ap5A and cooled on a bath of iced water for 2 min. The amount of ADP produced was quantified by the ADP-involved oxidation of NADH to NAD^+^ through the addition of 2.5 μL PK/LDH, 5 mM PEP, and 1 mM NADH at room temperature for 10 min. The enzymatic activity can be calculated from the absorbance at 340 nm measured with ultraviolet–visible spectroscopy (Agilent Technologies 8453/G1103A, Santa Clara, CA, USA). For each temperature, three independent experiments were performed with identical conditions simultaneously.

## 5. Conclusions

In conclusion, we developed a bioinformatics method using homologous primary sequences to improve the thermal stability of proteins and applied it to rationally redesign improved thermally stable variants of ADK wt. Our approach is directly applicable to any protein family with adequate thermophilic and mesophilic primary sequence information. Single- and double-point mutants constructed based on this method increase the thermal denaturation temperature of the mesophilic *E. coli* ADK by 5.5 and 8.3 °C, respectively, while preserving most of the catalytic function at ambient temperatures. The combination of our method with other theoretical methods is expected to provide more efficient guidance for stability modification in the future. In addition, our method might help to understand the intrinsic mechanism of species evolution when adapting to changing environmental conditions.

## Figures and Tables

**Figure 1 ijms-22-02768-f001:**
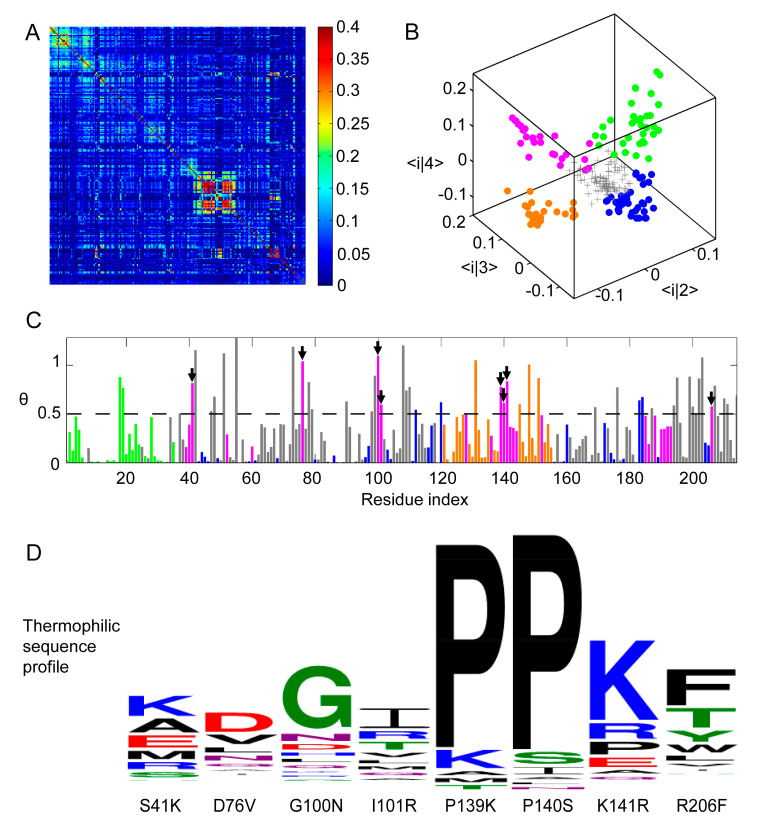
Residue Correlation Analysis (RCA) of Protein Sectors in ADK Family. (**A**) Heat map representation of RCA matrix *r_ij_* for MSA of ADK. *X**−* and *y*−axis coordinates both correspond to residue indices of ADK from *E. coli*, which has 214 residues. (**B**) Three dimensional scatter plot of the 214 residues in the space formed by the three eigenvectors of the second, the third, and the fourth largest eigenvectors. Each data point represented one position. After eliminating randomized background residues (gray), the rest of the positions could be clustered into four sectors, colored green, blue, orange, and magenta. (**C**) The relative entropy angle θ of thermophilic and mesophilic sequence profile at each position. Bars were colored by sectors. Residues whose angles were larger than 0.5 in the magenta sector were selected for further mutation (marked by arrow heads). (**D**) Amino acid distribution of thermophilic sequences at the chosen mutation positions. The first series of single-point mutations were S41K, D76V, G100N, I101R, P139K, P140S, K141R, and R206F.

**Figure 2 ijms-22-02768-f002:**
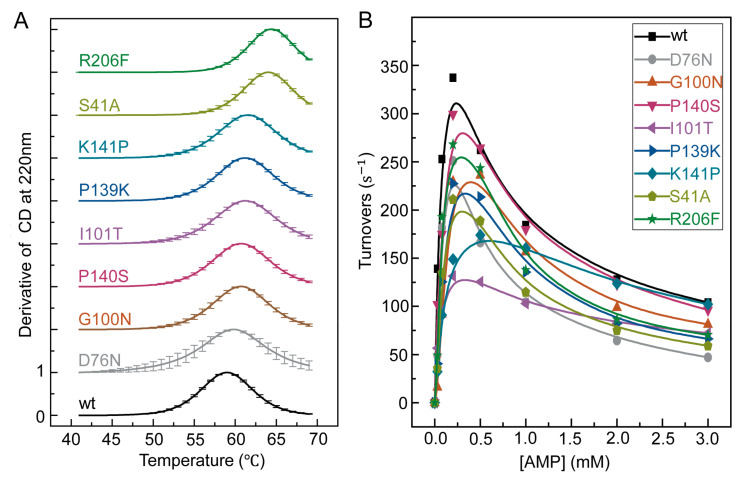
Thermal stability and enzymatic activity of single-point mutants characterized by circular dichroism (CD) spectroscopy and room temperature enzymatic activity assay. (**A**) CD differential signal at 222nm of wild type and mutants S41A, D76N, G100N, I101T, P139K, P140S, K141A, K141P, and R206F from 40 °C to 70 °C. All data were normalized to a scale of zero to one. To be distinguished from each other, each curve was offset by one along the *y*-axis. (**B**) Forward enzymatic activity of wild type and eight mutants at 25 °C. [ATP] = 1 mM, [magnesium acetate (MgOAc_2_)] = 2 mM.

**Figure 3 ijms-22-02768-f003:**
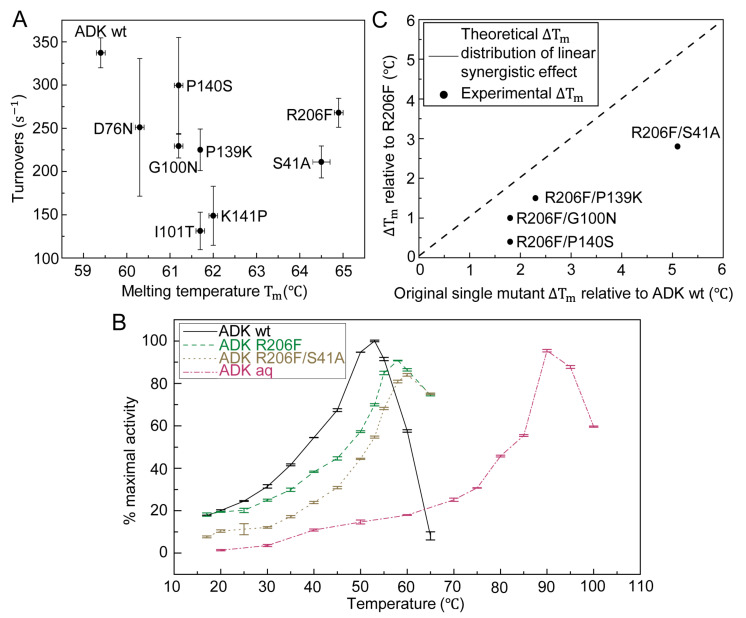
Correlation between melting temperature and enzymatic activity of wt and selected mutants of ADK. (**A**) Forward reaction activities (at 0.2 mM AMP and 1 mM ATP) versus melting temperature of ADK wild type and single mutants. (**B**) Temperature dependence of enzyme activity for ADK from *E. coli*, single mutant R206F, double mutant R206F/S41A and ADK from *Aquifex aeolicus* in the direction of ADP formation. The absolute value of 100% maximal activity was calculated as $458\pm 4\;s^{-1}$. (**C**) Four double mutants, R206F/S41A, R206F/G100N, R206F/P139K, R206F/P140S, all displayed a moderate *T_m_* increment lower than the sum of respective *T_m_* rise from two related individual mutants. The synergistic effect lowered as the Δ*T_m_* sum increased. The dash line indicated the theoretical Δ*T_m_* distribution of double mutants compared to R206F provided that the synergistic effect was linear.

**Figure 4 ijms-22-02768-f004:**
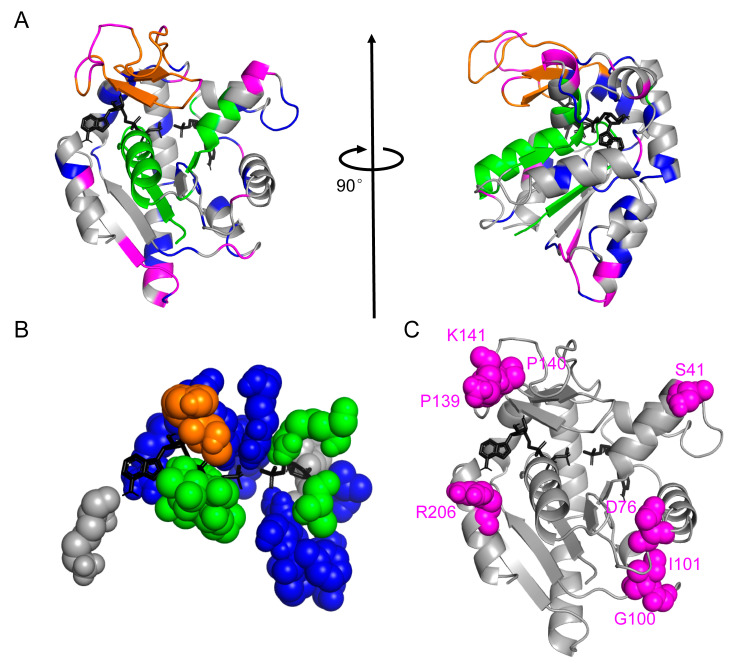
Mutations sites chosen by the relative entropy angle *θ*. (**A**) Residues colored according to our protein sectors color scheme on the “closed” conformation of ADK (PDB id: 1ANK chain A). ATP analog and AMP (both colored black) were shown in sticks. The structure on the right was obtained by rotating the diagram on the left by 90 degrees clockwise around the vertical axis. (**B**) Residues directly participating in substrate binding in *E. coli* ADK. All sectors except the magenta one were involved in ADK’s function execution. (**C**) Eight selected mutation sites were marked as magenta spheres in the tertiary structure of ADK.

**Table 1 ijms-22-02768-t001:** *T_m_* of ADK wt and all mutants, ranked in ascending order.

Protein	Melting Temperature (°C)
ADK wt	59.4 ± 0.1
D76N	60.3 ± 0.1
G100N	61.2 ± 0.1
P140S	61.2 ± 0.1
I101T	61.7 ± 0.1
P139K	61.7 ± 0.0
K141P	62.0 ± 0.1
S41A	64.5 ± 0.2
R206F	64.9 ± 0.1
R206F/P140S	65.3 ± 0.2
R206F/G100N	65.9 ± 0.2
R206F/P139K	66.4 ± 0.1
R206F/S41A	67.7 ± 0.3

## Data Availability

Data is contained within the article or Appendix A.

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
