# Peer review of "Rational Design of Adenylate Kinase Thermostability through Coevolution and Sequence Divergence Analysis"

_ijms, 2021, doi:10.3390/ijms22052768_

Round 1
Reviewer 1 Report
The manuscript entitled "Rational Design of Adenylate Kinase Thermostability through Coevolution and Sequence Divergence Analysis" by Jian Chang, Chengxin Zhang, Huaqiang Cheng and Yan-Wen Tan, describes a new method to design artificial mutant enzymes more thermostable than original wild one without loss of function, using E. coli adenylate kinase as a model enzyme. Based on their algorithm, they selected 8 residues out of 214, and experimentally substituted each to the candidate amino acid which may improves thermostability. The mutants showed increased melting temperature, retaining enzyme function. The results showed the protein design method is effective in improving thermostability of the enzyme.
The manuscript is written mostly in sound manner.
Several concerns are listed below.
(1) Evaluation of thermostability
Thermostability is measured using temperature dependent decomposition of secondary structure monitored by CD signal at 220 nm. The method is widely used for determination of melting temperature.
However, there is another criterion for enzyme thermostability, measuring time -dependent decrease of residual activity after incubating the enzyme at specified high temperature. This method reflects irreversible thermal denaturation of enzyme, different from the melting profile by CD signal.
The question is whether such irreversible thermal denaturation was tested using wild and mutant enzymes.
(2) Activity of mutants
L. 193-199; Double mutants such as R206F/I101T and R206F/K141P are not included. This may be because I101T and K141P did not "preserve better enzymatic activity while showing a larger Tm improvement" (L. 281). But these two mutants showed different behavior from others (Fig. 2B). Addition of some discussion about this point such as activation with Mg2+ is desirable.
In Fig. 2B, turnover is estimated at 0 - 3 mM AMP and 1mM ATP. Turnover at 1mM AMP is between 100-180 (s-1) for wild type and 8 mutants.
In Fig. 3A, turnover is estimated at 1 mM AMP and 1mM ATP. Turnover is between 100 - 340 (s-1) for wild type and 8 mutants. Especially the turnover of wild type is deviated. Legend to Fig. 3A (“1 mM” AMP) seems to be not correct but rather around “0.25 mM”.
Another question regarding this is as follows:
In the sentences in L. 189-191; “In the enzymatic forward activity assay at room temperature (Fig. 2B), these thermostable mutants had turnover rates at [adenosine triphosphate (ATP)] = [adenosine monophosphate (AMP)] = 1 mM that were 56%–98% of that of the wild type among which the most stable variant, R206F, retained 75% of the activity.”, R206 retained 75% of the activity.
However, in the sentences in L. 228-232; “We chose eight residues with the highest relative entropy angle θ values, namely S41, D76, G100, I101, P139, P140, K141, and R206, and constructed eight thermally stable single-point mutants. The maximum Tm increment of favorable single-point mutations can be as high as 5.5 °C. Meanwhile, the enzymatic activity can retain 75% of the performance at [ATP] = [AMP] = 1 mM compared with the wild-type ADK.” The 8 mutants, not only R206, retained 75% activity. This is overestimation.
(3) Activity at optimal temperature
Fig. 3B shows that R206F showed about 90% of activity compared with wild type, at their optimal temperatures. Since the optimal temperature of the reaction shifted, the evaluation of the activity by “Room Temperature Forward Activity Assay of ADK” (L. 464-479) seems not adequate to represent the enzyme activity. Some additional explanation is desirable.
(4) Structural insight in discussion is lacking.
All the 8 residues were on the peripheral location of the molecule. Two (G100, I101) and three (P139, P140, K141) are neighboring. Mutation of each position resulted in increase of melting temperature without significant loss of function. The latter may be explained by the fact that the catalytic center of ADK is not on the surface (the magenta sectors) of the molecule. But are the results about the increased stability explained in terms of three-dimensional structure -- such as interaction of side chain of mutated residues? What is the concrete meaning of “the stabilization effect of these residues on helices and sheets of ADK”(L. 243) ?
(5) Minor concerns;
L. 180; Please add "[ATP] = 1 mM, [MgOAc2] = 2 mM." so that the legend is comprehensive.
L. 217 (legend to Fig. 3B); Please describe the absolute value (micromole/min/mg enzyme or turnover) of 100% maximal activity.
Reviewer 2 Report
Chang et al, IJMS Review
This manuscript describes a novel method for enhancing thermostability of proteins. The authors use both multiple sequence alignment and evolutionary information from mesophile and thermophile amino acid sequences to design thermosatbilizing mutations to ADK. They then show experimentally that some of these mutations result in a higher melting temperature, only a small decrease in catalytic activity, and an increase in optimum catalysis temperature. Overall, this work is well executed and mostly well described. I have some issues with the manuscript which can be addressed by changes to the text. If these changes can be addressed, I consider the manuscript suitable for publication in IJMS without further experimental work.
Major concerns
The authors selected eight mutations to make (~ line 158). Some of the first round mutations resulted in a destabilization (lower Tm), but each of the second round mutations resulted in stabilization. The authors explained “the strategy for choosing mutation amino acid is by replacing the original residues by amino acids with apparently different properties”. However, I do not understand this logic. Three of the destabilizing first round mutations (S41K, D76V, I101R) already have different sidechain properties to the wild type sequence. Whilst the second round mutations (S41A, D76N, I101T, K141P) have different sidechain properties to the first round, this does not explain specifically why some of the mutations worked and others did not. It does not make sense that in all cases, the second round of mutations having “different properties” to the first round should work, whereas the first round having “different properties” to the wild type might not.
Related to the above point, the authors should present a more generalized discussion as to the potential limitations of this method. Although all mutations eventually chosen here resulted in thermostabilization, I would assume that this is not always the case. Under which situations might it work well, and not so well?
Line 198 “However, almost all double mutants showed further decrease in enzymatic activities compared with respective single mutants.” Please include the data for this statement, I don’t see it anywhere in the manuscript. (Even just a table, the actual curves can go in supplementary.)
Figure 2B – the authors mention in the text that the relevant turnover rates are at 1 mM ATP and 1 mM AMP, yet this figure shows a range of AMP concentrations, which is confusing and not really relevant to the text. Am I just meant to read off the 1 mM points from this graph? If so, then it would be easier to summarize the 1 mM data in the text and put the full AMP titration into supplementary materials.
Line 252 – 260 are new data. This should be in results, not discussion.
Lines ~280-290: these results were already discussed. The rest of the discussion after line 290 is good.
Minor concerns
In addition to the main points raised above, these specific issues could be improved upon:
Line 127: 256 and 3535 proteins. Presumably this refers to ADK homologous sequences, but this isn’t clear. Please clarify. And why doesn’t this add up to the 9203 sequences used in the initial analysis?
Line 181: A comment on the actual reaction catalyzed by ADK would be helpful for readers unfamiliar with the topic (i.e. ATP + AMP -> 2 ADP), somewhere around line 181.
Line 205: Why are these experiments now at 2.5 mM ATP and AMP rather than 1 mM for the previous set?
